# Cannabinoid-Induced Immunogenic Cell Death of Colorectal Cancer Cells Through De Novo Synthesis of Ceramide Is Partially Mediated by CB2 Receptor

**DOI:** 10.3390/cancers16233973

**Published:** 2024-11-27

**Authors:** Jeremy A. Hengst, Victor J. Ruiz-Velasco, Wesley M. Raup-Konsavage, Kent E. Vrana, Jong K. Yun

**Affiliations:** 1Department of Pediatrics, Pennsylvania State University College of Medicine, 500 University Drive, Hershey, PA 17033, USA; jah256@psu.edu; 2Department of Anesthesiology, Pennsylvania State University College of Medicine, 500 University Drive, Hershey, PA 17033, USA; vruizvelasco@pennstatehealth.psu.edu; 3Department of Pharmacology, Pennsylvania State University College of Medicine, 500 University Drive, Hershey, PA 17033, USA; wkonsavage@pennstatehealth.psu.edu (W.M.R.-K.); kvrana@pennstatehealth.psu.edu (K.E.V.)

**Keywords:** sphingolipids, ceramide, sphingosine phosphate, cell signaling, inflammation, immunogenic cell death, colorectal cancer, cannabinoids, (±) 5-epi CP 55,940

## Abstract

Mounting evidence demonstrates that cannabinoids have anti-cancer properties. The mechanism by which the cannabinoids induce cell death is still unclear. However, increased intracellular production of the sphingolipid, ceramide, seems to be a commonality. We recently demonstrated that a synthetic cannabinoid induced a specialized form of cell death that is known to activate the patient’s immune system, termed immunogenic cell death (ICD). Herein, we provide evidence of the mechanism by which synthetic cannabinoids increase ceramide production and demonstrate that ceramide is required for ICD. These findings strengthen the evidence that cannabinoids are effective anti-cancer agents and, importantly, suggest that they may help to recruit the immune system to fight the patient’s tumor.

## 1. Introduction

The dogmatic view that cancer chemotherapeutics induce apoptotic cell death, a non-immune activating or “tolerogenic” form of cell death, is being challenged by the recognition that certain chemotherapeutic strategies induce forms of cell death that are “immunogenic” (i.e., stimulate the immune system) [1,2]. The mechanism(s) by which certain agents induce “immunogenic” cell death (ICD) are only starting to be uncovered. Studies have demonstrated that cancer cells exposed to ICD-inducing agents, release adjuvants, termed danger-associated molecular patterns (DAMPs) [3]. These include the plasma membrane exposure of normally ER-resident chaperones such as calreticulin (ectoCRT), the autophagy-mediated secretion of ATP to the extracellular milieu, and the extracellular release of high-mobility group box 1 (HMGB1) [4]. One commonality among ICD-inducing agents is the induction of ER stress leading to phosphorylation/inhibition of eukaryotic initiating factor 2α (eIF2α Ser51). In fact, p-eIF2α has been regarded as “pathognomonic” to the induction of ICD [5].

DAMPs act as “find-me” and “eat-me” signals to recruit/activate the immune system to attack the tumor [6]. It is known that dendritic cells (DCs), “professional” antigen-presenting cells, express several pattern recognition receptors that act as sensors for the DAMPs exposed by dying cancer cells. These DCs initiate an immune response by presenting captured antigens and priming naïve T cells in lymphoid tissues to mount an attack on the cancer cells [6].

Recently, we were the first group to identify alterations to the sphingolipid metabolic pathway, including accumulation of ceramide (Cer) species, as an integral component of the process of ICD in colorectal cancer (CRC) cells [7]. Our findings suggested that other agents known to induce Cer accumulation might also induce ICD. One such group of agents are the cannabinoids (CBs), including endocannabinoids, phytocannabinoids, and synthetic CBs. The CBs have been studied as adjuvants to anti-cancer therapies primarily for their anti-pain/anti-inflammatory effects. However, there is ample evidence that certain phytocannabinoids, endocannabinoid analogs, and synthetic cannabinoids can induce cell death of cancer cells including colorectal cancer (CRC) [8,9,10,11]. We recently identified the synthetic CB (±) 5-epi CP 55,940 (hereafter referred to as 5-epi) as a potent inducer of ICD in CRC cells [12].

The CBs are known to induce accumulation of Cer through stimulation of the de novo sphingolipid synthetic pathway [8,9,10,13,14,15]. In our recent study, we confirmed that 5-epi induces de novo Cer synthesis [12]. We further demonstrated that 5-epi induced the marker of ICD, cell surface exposure of the DAMP, calreticulin (ectoCRT), in CRC cells. Consistent with our previous studies on the effects of sphingolipids in ICD [7], we demonstrated that 5-epi induces ER stress and eIF2α Ser51 phosphorylation. Moreover, consistent with the partially elucidated mechanism of ICD, 5-epi induced Caspase 8 activation, through degradation of its endogenous inhibitor c-FLIP, cleavage of its substrate Bap31, activation of Caspases 3/7 and cleavage of PARP [16]. These effects of 5-epi are consistent with the induction of bona fide ICD.

It was our goal, in this study, to determine whether 5-epi induces ICD through engagement of the cannabinoid receptors (CBRs), to examine the CBR selectivity of 5-epi, to elucidate the mechanism(s) by which 5-epi induces de novo sphingolipid synthesis, and to characterize the sphingolipids essential for induction of ICD induced by 5-epi.

## 2. Materials and Methods

### 2.1. Reagents and Antibodies

(±) 5-epi CP 55,940, SR 144528, SR 141716 (Rimonabant), ML-193, SB-705498, AM-630, JWH-133, HU-308, PM-226, GW405853, L759633, WIN 55,212-2, Δ^9^-THC, CBD and Myriocin were purchased from Cayman Chemicals (Ann Arbor, MI, USA). Thapsigargin, BAPTA-AM, L-cycloSerine, and PF-543 were purchased from Selleckchem (Houston, TX, USA). All compounds were prepared in a DMSO vehicle. Polyclonal CB1 and CB2 antibodies were obtained from Cayman Chemicals (Ann Arbor, MI, USA). Anti-phospho eIF2α Ser51 (D9G8) was purchased from Cell Signaling Technologies (Beverly, MA, USA). Anti-GAPDH (A-3) and Lass2 (CerS2; C-11) antibodies were obtained from Santa Cruz Biotechnologies (Dallas, TX, USA). Polyclonal Lass4 (CerS4) antibodies were purchased from ThermoFisher (Waltham, MA, USA). Polyclonal ORMDL3 antibodies were purchased from EMD Millipore (Billerica, MA, USA). All antibodies were used at 1:1000 dilutions, except for GAPDH which was used at 1:5000.

### 2.2. Cell Lines and Culture Conditions

Human HEK293 (CRL-1573), DLD-1 (CCL-221), HeLa (CRM-CCL-2), DU 145 (HTB-81), LN-229 (CRL-2611) and SH-SY-5Y (CRL-2266) cells were obtained from ATCC, (Manassas, VA, USA). SF-295 (SF001) cells were purchased from Neuromics (Edina, MN, USA). UACC903 cells were a kind gift from Dr. Gavin Robertson and 50B11 cells were provided by Dr. Victor Ruiz-Velasco, both of Penn State College of Medicine. All cells were cultured at 37 °C in a humidified atmosphere of 5% CO_2_ in Dulbecco’s Modified Eagle Medium (DMEM) supplemented with 10% fetal bovine serum (FBS) and penicillin/streptomycin.

### 2.3. Detection of Cell Surface Calreticulin (CRT)

For treatment, cells were seeded at 3 × 10^5^ cells/well in six well plates in 3 mL of DMEM with 10% FBS for 24 h, then transferred to DMEM containing 5% FBS and penicillin/streptomycin in the presence of treatments for 48 h. Cells were collected by trypsinization, followed by three washes in PBS containing 2% FBS, stained with anti-calreticulin (D3E6) phycoerythrin (PE) conjugated antibody (Cell Signaling, Beverly, MA, USA) at 4 °C for 1 h, washed as above, and CRT was detected using the Muse cell analyzer as previously described [12].

### 2.4. Sphingolipid Analysis

DLD-1 cells were treated with 5-epi (7.5 and 15 µM) and/or PF-543 (5 μM) for 48 h. Cells were collected by trypsinization, pelleted and washed with PBS and flash frozen. Sphingolipidomic analysis was conducted by the Lipidomic Shared Resource Facility (Medical University of South Carolina, Charleston, SC, USA). Sphingolipid levels were reported as pmoles of sphingolipid per nmole of inorganic phosphate (pmoles/nmole Pi) and were normalized to an average fold change relative to the vehicle treated controls.

### 2.5. Whole Cell Lysate Preparation

Total cell lysate was obtained by incubating treated and untreated DLD-1 cells in 1X RIPA buffer (Cell Signaling Tech, Beverly, MA, USA), with phosphatase inhibitor cocktail and Roche cOmplete protease inhibitor tablets (Millpore Sigma, Bedford MA), for 30 min at 4 °C and followed by removal of cell debris by centrifugation at 20,000× *g* at 4 °C. Protein concentrations were determined by BCA Assay (Pierce, Waltham, MA, USA).

### 2.6. Immunoblot Analysis

Protein samples were separated on 4–12% NuPAGE gradient gels and transferred to PVDF membranes under reducing or non-reducing conditions as noted. Membranes were blocked with 5% milk in TBS-T followed by incubation in primary antibodies for 1 h. After three 15 min-washes with TBS-T, membranes were incubated in respective secondary antibodies and visualized on X-ray film using Super-Signal West Dura reagents (Pierce, Waltham, MA, USA).

### 2.7. Statistical Analysis

Where appropriate, statistical analysis was performed using one-way ANOVA followed by a Tukey’s multiple comparison test using GraphPad PRISM (10.3.1). Results are reported as average values among replicates with standard deviation.

## 3. Results

### 5-Epi Induces ICD Through Selective Engagement of Cannabinoid Receptor 2 (CB2)

Our previous studies of the role(s) of sphingolipids in ICD have focused on colorectal cancer (CRC) cell model systems. However, the anticancer effects of CBs have been reported in multiple cancer cell types. Given that we would not expect that 5-epi induced ICD to be specific to a particular type of cancer, we collected a library of various human cancer cell types and normal cell lines and examined the production of a well-accepted DAMP and marker for ICD, cell surface exposure of normally ER-localized calreticulin (ectoCRT), as previously described [7,12,17].

Consistent with our previous results, 5-epi induces the production of ectoCRT in DLD-1 CRC cells treated for 48 h (Figure 1A). Similarly, 5-epi significantly induced ectoCRT exposure to varying degrees in five out of the seven cancer cell lines (DLD-1—CRC, HeLa—cervical, UACC903—melanoma, DU-145—prostate, and SH-SY-5Y neuroblastoma) examined regardless of their tissue of origin. Encouragingly, 5-epi minimally induced ectoCRT exposure in normal human (HEK293) and rat (50B11) cell lines. Interestingly, 5-epi did not induce ectoCRT in glioblastoma (GBM; SF295 and LN229) cell lines although the phyto-CBs (e.g., Δ^9^-THC and CBD) have been evaluated as potential anti-GBM therapeutics (recently reviewed by Buchalska et al.) [18]. Together, these results indicate that induction of ectoCRT exposure, by 5-epi, is not restricted to CRC, or to the DLD-1 human CRC cell line. Given that our previous studies have focused on CRC, we have chosen to further dissect the mechanism-of-action of 5-epi using DLD-1 cells as a model system.

The complexity of CB signaling regarding its receptors has been revealed through multiple studies. There are at least five well-established receptors for CBs and numerous “selective” CB agonists/antagonists have affinity with more than one receptor. For instance, Δ^9^-THC, the principle psychoactive component of cannabis, is an agonist of both CB1 and CB2 [19]. Cannabidiol (CBD), on the other hand, is an antagonist of CB1 and CB2 and therefore has no psychoactive effects [19]. CP 55,940, the parent compound of 5-epi, is a full agonist of both cannabinoid receptors (CB1 and CB2 [20]); thus, we considered the possibility that 5-epi could induce ICD through either or both of the canonical CBRs, or possibly through the non-canonical CB receptors. We therefore examined the relative expression levels of the canonical cannabinoid receptors, CB1 and CB2 in the cell lines tested above. As shown in Figure 1B, while there is variability in CB1 expression levels among the cell lines examined, CB2 receptor expression is relatively consistent. The response of the individual cell lines to 5-epi does not correlate with the relative expression level of either CB1 or CB2. Thus, we next attempted to discern the receptor selectivity of 5-epi, employing selective antagonists of the canonical and non-canonical CB receptors.

The CB2 inverse agonist, SR 144528 (10 μM), significantly reduced 5-epi induced ectoCRT exposure. Similarly, the CB1 inverse agonist, Rimonabant (SR 141716A), had modest, yet significant, effects on ectoCRT exposure at the same dose. Previous studies have demonstrated that SR 144528 is selective for the CB2 compared to other GPCRs at concentrations up to 10 μM and that SR 141716A is selective for CB1 at concentrations > 1 μM [21,22]. Antagonists of GPR55 (ML-193) and TRPV1 (SB-705498), other cell surface receptors that bind CBs, had no effect on ectoCRT exposure (Figure 1C). Together, these results indicate that 5-epi induces the production of ectoCRT via agonism of CB2 and, possibly, CB1.

To examine the relative contributions of CB1 and CB2 to the production of ectoCRT, by 5-epi, we next determined the dose-dependent effects of SR144528 and Rimonabant in DLD-1 cells. SR 144528 treatment, at 5 and 10 micromolar, resulted in a 40–65% reduction in ectoCRT exposure in DLD-1 cells treated with 5-epi for 48 h, respectively (Figure 1D). Under the same treatment conditions, Rimonabant attenuated ectoCRT production by 17–25%. We therefore tested another selective inverse agonist of CB2, AM630, and obtained results similar to those for SR144528 (Figure 1E). Thus, although CB1 antagonism may contribute to the reduction in ectoCRT exposure, it appears that CB2 is the primary target of 5-epi responsible for ectoCRT exposure/ICD induction.

In our earlier cannabinoid library screen, we demonstrated that 5-epi induces cytotoxicity of a panel of CRC cell lines [11]. We also identified several selective CB2 agonists that are not cytotoxic, at doses below 30 µM, including HU-308 and PM-226. One possible explanation for the discrepancy between 5-epi and the non-cytotoxic CB2 agonists would be that they have different functional effects on the CBRs. For instance, CB2 agonists can be biased toward the recruitment of β-arrestin, toward the inhibition of adenyl cyclase activity, or non-biased showing approximately equal effects on both signaling nodes [23].

To address this possibility, we chose several model agonists with varying selectivity toward adenyl cyclase inhibition. As shown in Figure 1F, in contrast to 5-epi, none of the tested agonists were able to induce ectoCRT exposure alone. However, four of the seven agonists tested (JWH-133, HU 308, PM 226, and GW405853) enhanced 5-epi-induced ectoCRT exposure, whereas L759633, Δ^9^-THC, and WIN 55,212-2 did not enhance 5-epi induced ectoCRT exposure. We obtained similar results with HU 308, PM 226, and L759633 in the mouse CRC cells lines CT-26 and MC38.

Based on the CB receptor bias studies of Dhopeshwarkar and Mackie, there is no clear correlation between the ability to enhance the effects of 5-epi and the functional bias of the agonists toward either β-arrestin recruitment or adenyl cyclase inhibition. These data demonstrate that CB2 engagement is involved in the induction of ectoCRT exposure. However, they also suggest that 5-epi might have another unknown effect on the CB2 that certain other agonists are able to amplify perhaps because they possess a weak ability to stimulate this unknown activity.

ER Stress and ER luminal Ca^2+^ depletion are required for 5-epi-induced ectoCRT exposure. ER stress and calcium depletion from the ER have been linked to the induction of ICD by the classical ICD inducer, mitoxantrone [16]. ER stress typically induces the unfolded protein response (UPR). Activation of one of the three arms of the UPR, protein kinase R-like ER kinase (PERK) and phosphorylation of eIF2α at Ser51, which inhibits cap-dependent protein translation, have been described as pathognomonic processes in ICD [5]. We previously demonstrated that 5-epi induces ER stress based on the enhanced phosphorylation/inactivation of eIF2α (Ser51) [12]. As shown in Figure 2A, in contrast to the effects of cytotoxic/ICD-inducing 5-epi, the non-cytotoxic CBs do not induce ER stress/phosphorylation of eIF2α. To determine if eIF2α was required for 5-epi-induced ectoCRT production, we over-expressed the dominant-negative eIF2α S51A mutant in DLD-1 cells. Inhibition of eIF2α phosphorylation/inactivation significantly reduced 5-epi-induced ectoCRT exposure (Figure 2B).

ER stress is associated with depletion of Ca^2+^ from its storage site in the ER and increases in cytosolic Ca^2+^ levels [16]. To determine if 5-epi induced ER Ca^2+^ depletion, DLD-1 cells were treated with the SERCA pump inhibitor, thapsigargin (Figure 2C), and the membrane-permeable cytosolic Ca^2+^ chelator, BAPTA-AM. Raising cytosolic Ca^2+^ levels/depletion of ER Ca^2+^ significantly enhanced the 5-epi-induced cell surface exposure of ectoCRT (Figure 2C), whereas lowering cytosolic/ER Ca2^+^ levels through intracellular chelation significantly reduced cell surface ectoCRT exposure (Figure 2D). These data are consistent with induction of ER stress and ER Ca^2+^ depletion in response to 5-epi treatment leading to ectoCRT and ICD.

5-epi induces de novo synthesis of Cer through alteration of Serine Palmitoyl-Transferase (SPT) regulation. The mechanism(s) by which CBs induce cell death have been partially delineated and increased de novo synthesis of the sphingolipid, ceramide (Cer), has been identified as a common occurrence [24]. We have previously demonstrated that 5-epi induces de novo synthesis of Cer [12]. Moreover, we demonstrated that inhibition of sphingosine kinase 1 (SphK1), which further increases flux through the de novo synthesis pathway, enhanced 5-epi-induced ectoCRT production. However, to date, whether Cer synthesis is required for cannabinoid-induced ICD, and the actual mechanism by which 5-epi and other CBs induces Cer synthesis, have not been elucidated.

Cer levels are tightly regulated in cells due to the inherent nature of Cer to induce cell death. Nevertheless, Cer is a required component of cellular membranes and is important for lipid raft formation and extracellular signaling cascades. Thus, the cell has developed a complex system to maintain homeostatic levels of Cer through regulation of the de novo sphingolipid synthesis pathway (Figure 3A).

Serine palmitoyltransferase (SPT) activity is the first and rate-limiting step in the de novo synthesis pathway. As the name implies, SPT catalyzes the combination of serine and palmitoyl-CoA to form 3-ketosphinganine. Inhibitors of the binding site for both serine and palmitoyl CoA have been identified and demonstrated to attenuate sphingolipid production. As shown in Figure 3B, competitive inhibition of palmitoyl CoA binding, using myriocin (Myr), and serine binding, using L-cycloSerine (L-cySer), both block the 5-epi-induced cell surface exposure of ectoCRT. These agents also attenuate ectoCRT production in response to CBD (Figure 3C) demonstrating that de novo synthesis of Cer is involved in CB-induced cell death, in general, and CB-induced ICD specifically.

SPT activity is regulated by a family of small, ER-resident, membrane-bound proteins, the orosomucoid-like proteins (ORMDL1-3) [25,26]. Importantly, Cer, itself, can bind to the ORDML proteins and enhance the interaction of ORMDL proteins with SPT [27]. Thus, Cer acts as a feedback sensor to regulate levels of its own synthesis (Figure 3A).

Given the multiple levels of regulation, it stands to reason that, under normal circumstances, activation of de novo synthesis of Cer should not lead to the induction of cell death. Our findings suggest that regulation of the de novo pathway is highly dysregulated by CBs. Considering that SPT activity is primarily regulated by the ORDML proteins, we examined the effects of 5-epi on protein expression of one of the three ORDML proteins (ORMDL3). As shown in Figure 3D, consistent with the stimulation of de novo sphingolipid synthesis, 5-epi dose-dependently reduced the expression of ORMDL3 by approximately 15% at 5 µM and 60% at 7.5 µM.

Cell death induced by 5-epi occurs after 24 h and reaches its plateau at 48 h. Thus, ORMDL3 depletion should occur at time-points preceding the appearance of cell death. As shown in Figure 3E, 5-epi alone induces the depletion of ORMDL3 as early as 12 h after treatment. Consistent with its ability to enhance 5-epi-induced production of ectoCRT [12], the small molecule SphK1 inhibitor, PF-543, enhanced the depletion of ORMDL3 at the 12 and 16 h time-points. Together, these findings indicate that 5-epi-induced ectoCRT production is mediated by Cer accumulation through enhanced de novo synthesis. Importantly, the dramatic increase in de novo synthesis, that we previously observed, is likely due to aberrant regulation of SPT activity mediated by depletion of ORMDL proteins.

Ceramide synthase 4 (CerS4) predominantly mediates 5-epi-induced ectoCRT exposure. The data presented above indicate that Cer accumulation is required for 5-epi-induced ectoCRT production. However, Cer encompasses numerous individual species comprised of variable acyl-chain lengths produced by one of six Cer synthases (CerS1-6) as detailed in Figure 4A. To determine which CerS enzyme(s) and acyl-chain length Cer species are involved in 5-epi-induced ectoCRT production, we separately stably over-expressed each of the six CerS in DLD-1 cells and examined the individual cell lines response to 5-epi and CBD (Figure 4B). Only over-expression of CerS4 led to significantly increased ectoCRT production in response to both CBs. CerS4 has a substrate preference for C18 and C20 acyl-chain CoAs, while CerS2 favors C22 and C24-CoAs but can also utilize C20-CoA. Thus, these two over-expression cell lines were chosen for further examination.

Interestingly, there appears to be a reciprocal enhancement of CerS2 in CerS4 over-expressing cells and vice versa in CerS2 over-expressing cells (Figure 4C). Compared to wild-type DLD-1 cells, only CerS4 and to a lesser extent CerS2 over-expression resulted in increased cell surface exposure of ectoCRT (Figure 4D). At a 3 µM dose, 5-epi alone does not induce ectoCRT production, nor does over-expression of CerS2, whereas CerS4 over-expression appears to be sufficient for enhancement of ectoCRT production suggesting that C20 Cer species are responsible for the observed effects (Figure 4D). Together, these results over a range of 5-epi and CBD doses increase the likelihood that CerS4 is primarily responsible for cell surface exposure of ectoCRT.

5-epi-induced activation of de novo Cer synthesis induces a dramatic increase in C20:4 Cer. We previously reported the effects of 5-epi on sphingolipid levels in CRC cells and determined that, in general, Cer species were upregulated by an average of 5- to 10-fold [12]. However, our initial characterization focused on the most predominant Cer species (C16:0, C18:0 and C24:1). Thus, we have re-analyzed the entire Cer species dataset and this analysis revealed a very dramatic and selective accumulation of one Cer species, C20:4 Cer (Figure 4E). Indeed, C20:4 Cer levels were increased 11-fold at 7.5 µM 5-epi and 21-fold at 15 µM 5-epi as compared to vehicle treatment. Remarkably, the combination of 7.5 µM 5-epi + PF-543 resulted in a >60-fold increase in C20:4 Cer levels. None of the other Cer species increased more than 10-fold with 5-epi alone or in the presence of PF-543 [12].

C20:4 Cer is generated, in the de novo pathway, by the condensation of dhSph (d18:0) and arachidonoyl-CoA (C20:4) to form C20:4 dhCer/Cer. However, CerS2/4 also generate C20:0 and C20:1 dhCer/Cer using their respective acyl-CoAs [28]. Importantly, C20:0 and C20:1 Cer levels do not increase to the same extent as C20:4 (Figure 4E) indicating that CerS2/4 activity is not specifically activated by 5-epi treatment but, rather, that arachidonoyl-CoA (C20:4 CoA) levels have dramatically increased relative to other acyl-CoAs.

## 4. Discussion

Taken together, these findings expand our understanding of the mechanism by which (±) 5-epi CP 55,940 (and by inference other CBs) induces immunogenic cell death of CRC cells. 5-epi functions, at least in part, through engagement of the CB2, and induces ER stress and ER Ca^2+^ depletion leading to de novo synthesis of Cer species. Moreover, we demonstrate, for the first time, that aberrant de novo synthesis of Cer occurs through depletion of the endogenous regulators of the de novo synthesis pathway—the ORMDL proteins. Importantly, we also demonstrate that CerS4 activity is directly involved in the production of ectoCRT and that 5-epi exerts a previously unrecognized effect on the intracellular levels and/or metabolism of arachidonic acid.

Our findings indicate that the ability to induce ICD is not unique to 5-epi, as CBD also induces the cell surface exposure of ectoCRT (Figure 3C). Whether 5-epi and CBD induce ectoCRT exposure through similar mechanisms requires further investigation. Nor is the ability of CBs to induce Cer synthesis unique to these agents. For instance, Gustafsson et al. reported that an analog of the endocannabinoid anandamide, R-methanandamide, induced Cer synthesis and cell death. Moreover, the effects of R-methanandamide were enhanced by inhibition of SphK1 and glucosylceramide synthase, similar to the effects of the SphK1 inhibitor, PF-543, observed here and previously [15]. Interestingly however, R-methanandamide appears to predominantly engage CB1, whereas 5-epi predominantly works through the CB2. There were also differences in the CerS enzymes associated with cell death. Gustafsson and colleagues identified CerS3 and CerS6 and C16 and C24:1 Cer as required for R-methanandamide-induced cell death. In this study, we identified CerS2 and CerS4 as important for 5-epi-induced ICD. It is interesting to speculate that CB1 and CB2 have differential effects or cell-specific pathway influences that lead to differential changes in Cer accumulation. It will also be interesting to determine whether R-methanandamide induces ICD or “tolerogenic” apoptotic cell death.

One key observation from this study is that 5-epi induces, and SphK1 inhibition enhances, the depletion/breakdown of ORMDL3. Similar reductions in the other ORDML proteins, ORMDL1 and 2, are possible and could therefore significantly reduce the constitutive inhibition of SPT activity. Moreover, if ORMDL protein expression is significantly reduced, Cer is no longer capable of negatively regulating SPT activity through a feedback inhibition mechanism mediated by ORMDL: Cer interactions. Thus, SPT activity is uninhibited leading to dysregulated de novo synthesis of Cer.

In addition to the regulation of SPT activity, another level of regulation of Cer levels in the de novo pathway was revealed by Siow et al. [25]. They demonstrated that SphK1 converted the majority of the dhSph, generated in the de novo pathway, to dhS1P which was subsequently irreversibly hydrolyzed by S1P lyase (S1PL; Figure 3A). This activity of SphK1/S1PL severely limits the availability of dhSph for dhCer production, making dhSph a limiting factor for the de novo Cer production. Indeed, this is what we previously observed where SphK1 inhibition dramatically enhanced the levels of dhSph produced in response to 5-epi treatment [12].

However, the observation that SphK1 inhibition enhances ORMDL3 depletion indicates that S1P may also play a role in the stabilization of ORDML proteins. We have recently observed that S1P stabilizes the endogenous inhibitor of Caspase 8, c-FLIP [17]. S1P has been reported to function as a required co-factor for the E3 ubiquitin ligase activity of TRAF2 [29]. TRAF2 catalyzes the polyubiquitination of target proteins through conjugation of ubiquitin monomers linked together at the lysine 63 (K63) position of the ubiquitin monomer. Unlike K48-linked polyubiquitination, which is associated with protein degradation, K63-linked polyubiquitination regulates signaling pathways by increasing protein stability [30]. We believe that S1P-mediated K63-ubquitination stabilizes these proteins by protecting them from proteasomal breakdown. SphK1 inhibition reduces intracellular S1P, thereby attenuating K63-Ub ligase activity of TRAF2, or others, and enhances the catalytic turnover of substrates such as c-FLIP, and possibly the ORMDL proteins.

Although we previously observed that SphK1 inhibition enhanced dhSph levels in response to 5-epi, we did not observe a further increase in Cer levels among the most abundant and frequently examined Cer species [12]. However, in this study, we observed a dramatic increase in C20:4 Cer levels. As these data are derived from the same study set, it is possible to make the direct correlation that, in the presence of saturated levels of dhSph, there is an unusual increase in free arachidonic acid (AA; C20:4) in response to CBs, whereas the other acyl chain lipid species such as palmitic acid, stearic acid, etc., remain unchanged and thus, rate-limiting for Cer synthesis.

We have considered multiple intracellular sources of this AA pool including the catabolic breakdown of the endoCBs, 2-arachidonylglycerol, and anandamide. AA can also be liberated from phospholipids and plasmalogens by the action of cytosolic phospholipase A2 (cPLA2) [31,32]. Interestingly, Cer-1-phosphate (C1P), the product of Cer kinase (CerK)-mediated phosphorylation of Cer species, is a known stimulator of cPLA2 activity [31,32]. Given the increased production of Cer by CB stimulation of de novo Cer synthesis, it is possible that C1P production is also elevated in CRC cells in response to CBs. C1P-stimulated cPLA2 activity could liberate AA from the *sn-2* position of plasmalogens, further increasing the available ER pool of AA. Given the increased availability of AA (phospholipid/plasmalogen breakdown) and dhSph (de novo synthesis) at the ER, CerS2/4 utilizes these two substrates to form C20:4 Cer. Interestingly, the *sn-2* acyl chains of plasmalogens are predominantly docosahexaenoic acid (DHA, 22:6) and AA (20:4). Typically, sphingolipidomic analyses do not measure DHA incorporation in Cer, but it is possible that C22:6 Cer could be formed and drastically increase in response to combined CB and SphK inhibitor treatment.

AA is the precursor of prostaglandin synthesis, including prostaglandin E2 (PGE2). CBs have been demonstrated to have anti-inflammatory and anti-pain activities. PGE2 has also been associated with inflammatory pain [33]. PGE2 has also recently been identified as an inhibitor of ICD or “inhibitory DAMP” (iDAMP) that counteracts the “eat-me” signal, ectoCRT [34]. Thus, the anecdotal evidence of an anti-cancer effect of CBs could be explained by their ability to simultaneously induce the production of “activating DAMPs” (e.g., ectoCRT) and inhibition of the production of the iDAMP, PGE2. Similarly, the better described anti-pain/anti-inflammatory effects of the CBs could be explained by their inhibition of PGE2 synthesis.

## 5. Conclusions

In conclusion, the studies conducted herein, along with our previous studies of 5-epi, provide evidence that (±) 5-epi CP 55,940 induces ICD through induction of de novo Cer synthesis. Moreover, they suggest that 5-epi is a synthetic cannabinoid and primarily induces the above effects through agonism of CB2. At this point, it is not possible to say whether 5-epi binds to both CB1 and CB2, or other receptors (CB or otherwise) given that the preparation of 5-epi we employed was a racemic mixture. Future studies to more finely dissect the identity of the receptors involved require chiral separation of the racemers, and molecular inhibition of the individual CB receptors. Importantly, these studies also shed light on the mechanism by which CBs induce de novo synthesis of sphingolipids. However, the exact mechanism by which CBs induce the downregulation of ORMDL3, and possibly other ORMDL proteins, is unknown at this time.

Moreover, our data suggest that 5-epi and potentially other CBs induce changes to the metabolism of arachidonic acid that are reflected by dramatic increases in C20:4 Cer. They further implicate CerS4 as the predominant enzymatic activity responsible for the increase in production of C20:4 Cer, consistent with the known substrate specificity of CerS4. Further studies are required to more precisely determine the source of the increase in arachidonic acid, how this pool becomes available for Cer synthesis, the Cer synthase(s) responsible for C20:4 Cer production, and whether C20:4 Cer is directly responsible for induction of ICD by CBs including, but not limited to 5-epi.

Lastly, our data indicate that inhibition of the sphingosine kinases can significantly increase cannabinoid-induced Cer accumulation. Our studies thus far have employed multiple SphK inhibitors at concentrations where both SphK1 and SphK2 are inhibited (e.g., 5 µM PF-543). We believe that both SphK1 and SphK2 must be inhibited to sufficiently reduce S1P production and enhance Cer production to levels sufficient for enhancement of 5-epi induced-ICD. However, further dissection of the roles of the individual SphKs through molecular inhibition would clarify this issue.

Interestingly, natural polyphenols including resveratrol (red wine) and epigallocatechin gallate (EGCG; green tea) have been shown to inhibit SphK activity in animal models [35]. Given that all of these agents are legally available over the counter and in widespread use among healthy people as well as cancer patients, it would be interesting to assess their efficacy as chemopreventatives or as adjuvants to standard-of-care chemotherapy. Thus, clinical trials to assess their single/combined ability to enhance the efficacy of standard-of-care agents that induce ICD should be expedited.

## Figures and Tables

**Figure 1 cancers-16-03973-f001:**
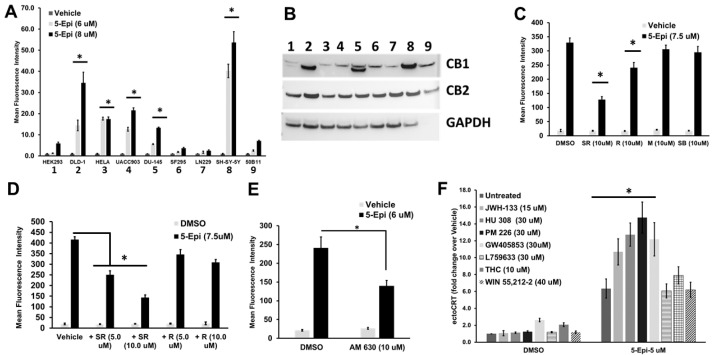
5-epi induced exposure of ectoCRT is partially mediated through CB2 agonism in CRC cells. (**A**) 5-epi dose dependently induced ectoCRT exposure in human CRC, cervical, melanoma, prostate, and neuroblastoma cell lines. Human glioblastoma and normal kidney epithelial, as well as normal rat cell lines did not significantly respond to 5-epi at the indicated concentrations. (* = *p* < 0.05; *n* = 3). (**B**) The relative expression levels of CB1 and CB2 were determined by Western blot analysis in the cell lines examined in (**A**). GAPDH was included as a loading control. The GAPDH antibody does not recognize the protein of rat origin. (**C**) 5-epi (7.5 µM) induced ectoCRT production was significantly attenuated by CB2 (SR; SR144528) and CB1 (R; SR141716) inverse agonists. (* = *p* < 0.05; *n* = 3) Antagonists of GPR55 (M; ML-193) and TRPV1 (SB; SB-705498) did not affect 5-epi-induced ectoCRT production. (**D**) Both SR and R dose dependently affected ectoCRT production. SR was more effective, implying a greater contribution of CB2 signaling to ectoCRT production. (* = *p* < 0.05; *n* = 3) (**E**) 5-epi induced ectoCRT exposure was significantly inhibited by a selective CB2 antagonist (AM 630). (* = *p* < 0.05; *n* = 3). (**F**) Among the CB2 agonists tested, only 5-epi (untreated) induced ectoCRT exposure in DLD-1 cells treated for 24 h. (* = *p* < 0.05; *n* = 3) JWH-133, HU-308, PM 226 and GW405853 enhanced 5-epi-induced ectoCRT production. (* = *p* < 0.05; *n* = 3), whereas L759633, THC, and WIN 55,212-2 had no effect on 5-epi-induced ectoCRT production.

**Figure 2 cancers-16-03973-f002:**
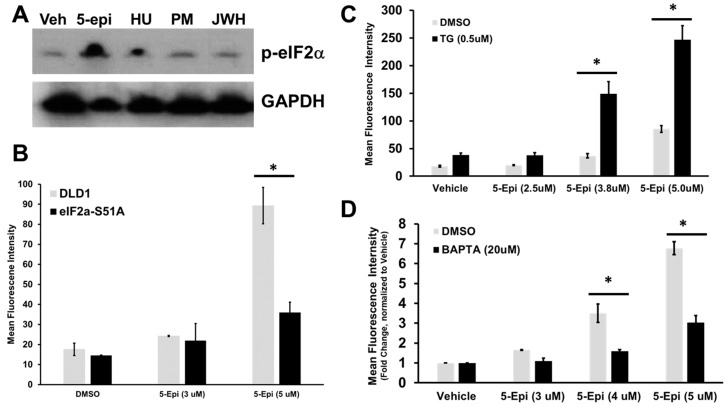
5-epi induces ER stress and ER Ca^2+^ depletion leading to ectoCRT exposure. (**A**) DLD-1 cells were treated with 5-epi and non-cytotoxic CBR agonists. ER stress was monitored by Western blot of phospho-eIF2α (Ser51). GAPDH was included as a loading control. (**B**) Expression of the dominant–negative eIF2αS51A mutant abrogates 5-epi induced ectoCRT exposure (*n* = 3). (**C**,**D**) EctoCRT exposure was determined after 24 h of treatment in the presence or absence of thapsigargin (**C**) and BAPTA-AM (**D**). (* = *p* < 0.05; *n* = 3).

**Figure 3 cancers-16-03973-f003:**
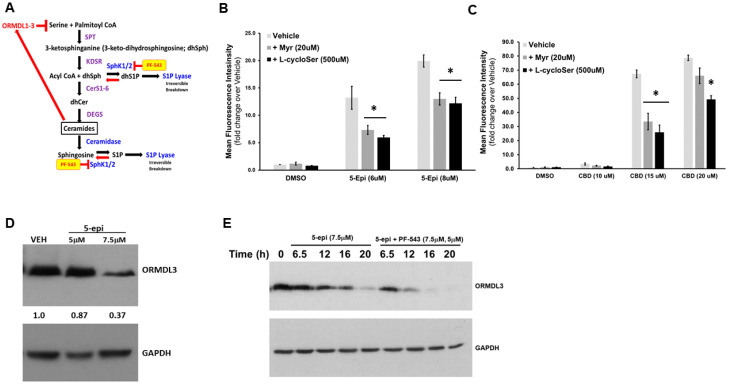
5-epi stimulates de novo synthesis of Cer through depletion of ORMDL3. (**A**) SPT activity, the first step in the de novo synthesis pathway, is regulated by a family of small, ER-resident, membrane-bound proteins, the orosomucoid-like proteins (ORMDL1-3). ORMDL proteins bind to and inhibit SPT activity. Importantly, Cer can bind to the ORDML proteins and enhance the interaction of ORMDL proteins with SPT. Thus, Cer acts as a feedback sensor to regulate levels of its own synthesis. (**B**) DLD-1 cells were treated with the indicated concentrations of 5-epi for 24 h in the presence and absence of the inhibitors of de novo Cer synthesis, myriocin (Myr), and L-cycloserine (L-cycloSer). EctoCRT production was determined by flow cytometry. (*n* = 3). (**C**) DLD-1 cells were treated with the indicated concentrations of CBD for 24 h in the presence and absence of the inhibitors of de novo Cer synthesis, myriocin (Myr) and L-cycloserine (L-cycloSer). EctoCRT production was determined by flow cytometry. (*n* = 3). (**D**) DLD-1 cells were treated with the indicated concentrations of 5-epi for 24 h. Levels of ORMDL3 and GAPDH were determined by Western blot analysis. (*n* = 3). (**E**) DLD-1 cells were treated with the indicated concentrations of 5-epi in the presence or absence of PF-543 at various time points. Levels of ORMDL3 and GAPDH were determined by Western blot analysis on separate blots. (*n* = 3) (* = *p* < 0.05).

**Figure 4 cancers-16-03973-f004:**
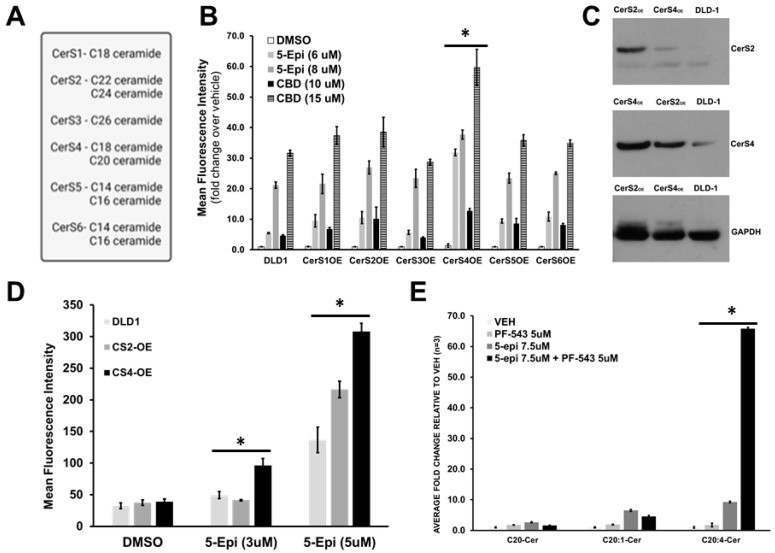
CerS4 activity mediates 5-epi induced production of ectoCRT and dramatic accumulation of C20:4 Cer in response to 5-epi + PF-543 treatment. (**A**) The predominant Cer species produced by each CerS (CerS1–CerS6) is listed. (**B**) DLD-1 cells over-expressing the indicated CerS enzyme were treated with the indicated concentrations of 5-epi or CBD for 24 h. EctoCRT production was determined by flow cytometry. (*n* = 3). (**C**) Wild-type DLD-1 cells and single clones over-expressing CerS2 or CerS4 were examined by Western blot analysis. (**D**).DLD-1 cells over-expressing CerS2 or CerS4 were treated with the indicated concentrations of 5-epi for 24 h. EctoCRT production was determined by flow cytometry. (*n* = 3). (**E**) DLD-1 cells were treated with the indicated concentrations of 5-epi in the presence or absence of PF-543 for 48 h. Levels of C20 Cer species were determined by LC/MS/MS analysis. (*n* = 3) (* = *p* < 0.05).

## Data Availability

The original contributions presented in this study are included in the article. Further inquiries can be directed to the corresponding author.

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
