# Peer review of "Cannabinoid-Induced Immunogenic Cell Death of Colorectal Cancer Cells Through De Novo Synthesis of Ceramide Is Partially Mediated by CB2 Receptor"

_cancers, 2024, doi:10.3390/cancers16233973_

Round 1
Reviewer 1 Report (Previous Reviewer 1)
Comments and Suggestions for Authors
The authors responded to the previous review mostly by refuting the points raised in the previous review.
They cite a reference that de novo synthesized ceramide has previously been implicated in cannabinoid induced cell death (DOI: 10.1042/0264-6021:3630183). While this publication did use l-cycloserine, they did not use myriocin. The authors then go on to justify the dose of myriocin used in the manuscript stating it is often used in the micromolar range, when in fact most in cell assays use 100nM. Similar issues are still present for PF-543.
Knockdowns for each of the CBRs, as well as SKs and FB1 to inhibit CerS should be included.
Author Response
Please see the attachment.

Reviewer 2 Report (Previous Reviewer 2)
Comments and Suggestions for Authors
Jeremy and colleagues have reported an interesting relationship between cannabinoid-mediated death of colorectal cancer through their titled manuscript, “Cannabinoid-induced immunogenic cell death of colorectal cancer cells through de novo synthesis of ceramide is partially mediated by CB2 receptor”. In this article, it was focussed on ceramide (Cer)-facilitated immunogenic cell death and proposing a mechanism of 5-epi-induced-Cer accumulation activated by CB2 receptors. Authors also stressed the importance of ceramide synthase 4, a key factor in immunogenic cell death (ICD). Authors proposed de novo sphingolipid synthesis pathway (Figure 3A). The work has been revised adequately to be publishable grade. The manuscript has all the components and figures and research supported by the literature mentioned in the introduction section. References are updated and results are discussed adequately.
However, minor corrections are to be addressed as mentioned below:-
1. Authors employed the racemic mixture (±) 5-epi CP 55,940. Is there any chiral resolution of the racemic mixture was conducted to separate isomers of this compound? It is imperative to know which isomer is more active and responsible for the induction of ICD. Since it is a synthetic cannabinoid, the chiral separation is quite possible. Authors should comment on this.
2.Line-117: six well plates or ninety-six well plates?
Author Response
Please see the attachment.

This manuscript is a resubmission of an earlier submission. The following is a list of the peer review reports and author responses from that submission.
Round 1
Reviewer 1 Report
Comments and Suggestions for Authors
The authors set out to extend their previous work demonstrating a role for sphingolipids in ICD. They demonstrate that the synthetic cannabinoid 5-epi, induces ICD in several cell lines and implicate de novo sphingolipid synthesis in this process. However, there are several significant weaknesses that need to be addressed in the manuscript.
Figure 1B demonstrates the roles for CB1 and CB2 in ectoCRT exposure using inverse agonists and antagonists. They conclude that "these results indicate that 5-epi induces the production of ectoCRT via agonism of CB2 and, possibly, CB1. This conclusions would be better supported with additional experiments using cells with knockdown or knockout of the CB receptors. Especially given the difference in response with 10uM SR141716 in Figure 1 B and C.
The authors use 20uM myriocin in Figure 3. This is an excessive and inappropriate dose of the SPT inhibitor, as it works in the nM range. These experiments should be repeated with appropriate doses to rule out off target effects. Similarly, L-cycloserine has been shown to inhibit generation of sphingolipids in the 10-50uM range. Data demonstrating that these inhibitors are decreasing sphingolipid generation are necessary to support their conclusion that "that de novo synthesis is required for CB-induced cell death." PF543 also inhibits SphK1 in the low nM range.
There are no statistics included in Figure 4B to support the authors statement that "Compared to wild-type DLD-1 cells, only CerS4 and to a lesser extent CerS2 over-expression resulted in increased cell surface exposure of ectoCRT. These results were consistently observed over a range of 5-epi and CBD doses increasing the likelihood that CerS4 and possibly CerS2, are responsible for cell surface exposure of ectoCRT (Figure 4B)." Moreover the data in CerS1, CerS2, CerS5 and Cers6 OE cells are nearly identical making the conclusion that CerS2 may be involved unlikely or hard to justify based on the data in the other overexpressing cell lines. In addition CerS4 expression is very low in CRC cells. Studies using knockdown or knockout for each of the CerS would be better suited to determine the role of specific CerS enzymes. These data are further over-concluded with the statement "However, at very low doses of 5-epi, CerS4 over-expression appears to be sufficient for enhancement of ectoCRT production suggesting that C20 Cer species are responsible for the observed effects (Figure 4D)." The data that are included are fold-change and absolute values should be included.
Reviewer 2 Report
Comments and Suggestions for Authors
Jeremy and co-workers presented an article titled, “Cannabinoid-induced immunogenic cell death of colorectal cancer cells through de novo synthesis of ceramide is partially mediated by CB2 receptor”. The authors demonstrated the relationship between cannabinoid (5-epi) induced ceramide (Cer) that subsequently facilitates immunogenic cell death in colorectal cancer cell lines. Further, the authors investigated the participating mechanism of 5-epi-induced-Cer accumulation through the involvement of CB2 receptors, via sphingolipids and found that ceramide synthase 4, a key factor is also required in the process of 5-epi-induced immunogenic cell death (ICD). The de novo sphingolipid synthesis pathway has been shown in Figure 3A is quite appreciable. The work is very interesting and advances the present knowledge to the related research fields.
This work is interesting and opens a new door to investigate the regulatory mechanisms involved in the de novo ceramide synthesis pathway.
The work is presented well and the literature covers all the basic information about the study in the introduction section.
References covered recent literature reports and are found adequate and complete.
Results are described well and the conclusion is in consonance to the aim of the study.
All the figures are appropriately designed and presented clearly.
Following minor revision is required:-
1Authors employed the racemic mixture (±) 5-epi CP 55,940. Is there any chiral resolution of the racemic mixture was conducted to separate isomers of this compound? It is imperative to know which isomer is more active and responsible for the induction of ICD. Since it is a synthetic cannabinoid, the chiral separation is quite possible. Authors should comment on this aspect.
2In Line 85, the authors mentioned bona fide ICD., correct the formatting/typo error.
3The statements mentioned in Lines 151 (anti-pain/anti-inflammatory effects.) and 152, (directly cytotoxic to cancer cells ), provide the published research reports, respectively.
Reviewer 3 Report
Comments and Suggestions for Authors
The study by Hengst et. al. is an informative study into the mechanism underlying cannabinoid-induced immunogenic cell death. The authors have previously shown that a synthetic cannabinoid 5-Epi induced immunogenic cell death as measured by expression of ectoCRT. In the present study they determined that the predominant receptor for this mechanism of action is cannabinoid receptor 2 (CB2), with some contribution of CB1, that appears to be selective to 5-epi. This effect is dependent on ER stress phosphorylation of EIF2alpha and depletion of ER calcium stores. The authors then went on to further characterize how the treatment with 5-epi impacts de novo sphingolipid synthesis which leads to cell death. They find that a likely increase in Serine Palmitoyl-Transferase activity due to depletion of its regulator ORMDL1-3 by 5-Epi treatment induces ceramide synthesis. Finally, they determined that the increase in C20:4 Cer, likely due to the activity of CerS4, is the key ceramide for ectoCRT increase. The study is mostly clear and well-written where the authors sufficiently explain their motivation behind each experiment and thus their hypotheses and their testability. For the most part, it is also well-controlled, especially in the later experiments, with some exceptions noted below under major concerns. The discussion is a strength of this paper as it considers multiple interpretations and ties their findings into the greater field very well. Overall, this study contributes to our understanding of cannabinoid induction of immunogenic cell death and upon addressing the specific concerns listed below will be a worthwhile addition to the literature on the topic.
Specific comments
Major concerns:
1) For transparency and reproducibility, the authors should list the specific clones of the antibodies used or state that the antibodies used are polyclonal. In any case, species should also be listed. Furthermore, the authors should also provide the dilutions used for the antibodies in appropriate protocols i.e. Immunoblot analysis what dilution of primary antibodies was used? Etc.
2) In Figure 1 the authors report that 5-Epi treatment of normal human (HEK293) and rat (50B11) cell lines minimally induced ectoCRT in response to 6uM or 8uM treatment vs. vehicle control which suggest non-cancer cells are not affected by the treatment indicating a good safety profile. Interestingly, the same treatment of glioblastoma lines with 5-Epi did not induce any ectoCRT despite CBs therapeutic potential in glioblastoma. It would be thus relevant to measure the expression of receptor CB2 in all the cell lines used to determine if the level/presence of CB2 receptor influences the degree of ectoCRT response to 5-Epi treatment. This is important and relevant as higher doses of 5-Epi clearly induce a slight increase in ectoCRT in HEK293 and 50B11 cell lines, suggesting higher concentration could induce more ectoCRT and thus at these higher doses ‘non-cancerous’ cells might still be undergoing ICD. I believe the tested doses of 5-epi were derived from “concentrations close to the reported IC50 in DLD-1 cell” as described in the previous paper by the group where they used 3.8uM, 5uM, and 7.5uM 5-Epi (Hengst, 2022, reference 11). Thus, these doses might not always induce same level of cell death for all cell lines tested here and non-cancerous cells might not always be “safe” from treatment based on drug concentration needed to target the specifcy type of cancer.
3) Connected to the previous point, an explanation for why these doses of 5-epi were tested across all the cell lines should be added.
4) The authors state that not all CB2 agonists were cytotoxic to a panel of CRC cell lines in their previous study. Do we know if 5-Epi is cytotoxic to all the cell lines in Figure 1A at the tested concentrations? If yes, please state and cite. If not, this would be relevant information to show in supplementary data for 5-Epi doses used.
5) In describing the evaluation of CBs in glioblastoma treatment (lines 166-168) relevant references need to be added at the end. Furthermore, the outcome of these “evaluation studies” should be described-is CB treatment effective in glioblastoma treatment in the clinic? The authors seem to imply it is which makes not seeing ICD induction in glioblastoma lines interesting, but the actual results from these studies should be definitively described in text.
6) Why are some results for ectoCRT shown as raw fluorescence while others are shown as fold-change over vehicle? This makes is difficult to compare results across different experiments so I suggest choosing a unifying approach unless there is a scientific reasoning behind the choice to show them one way one time and different way another time.
7) In lines 285-287 describing the results of inhibiting SPT in Figure 3, the authors state that “de novo synthesis [of sphingolipid Cer] is required for CB-induced cell death, in general, and CB-induced ICD specifically.” but without any cell death data presented, is it possible to conclude the cell death part?
8) Based on Figure 1C and 1D the authors can conclude that CB2 is the primary target of 5-Epi but that CB1 still has some contributions. This is commendably reflected in the title as “partially mediated by CB2 receptor” but the abstract language suggests only CB2 receptor is the target. It would be appropriate to add a qualifier here as done in the title.
Minor concerns:
1) In line 33 the authors state “alteration of intracellular calcium levels” but it is more informative to describe this as a calcium depletion rather than general phrase “alteration”.
2) In Line 49 the authors state “numerous studies have demonstrated” yet only cite a single review, thus it would be appropriate to either remove the word “numerous” or to cite additional studies showing the phenomena.
3) In line 113 the authors state the concentration of cells used, but for transparency and reproducibility, the volume should also be provided so that the total number of cells analyzed can be inferred.
4) Lines 149-153 would be more appropriate in the introduction before introducing their work on cannabinoids, maybe around line 70.
5) Figure 1 legend should contain description of panel A as bolded Figure Title is not entirely applicable to the data shown. Additionally, the legends in figure 1 are too small to discern patterns in the boxes for appropriate bar graphs. Please use color or increase the legend size to improve the quality of data presentation. Furthermore, I do not believe any SphK1 inhibition was done in experiments shown in Figure 1 so “and enhanced by 173 SphK1 inhibition” should be removed from line 173 if I am not mistaken.
6) In all figures, please indicate what comparison is the significance star referring to in all panels like you have indicated in Figure 1 panel A.
7) Bar graph results shown in all figures have error bars and the methods appropriately state that the results are reported as average values among replicates, but it is standard practice to report how many independent replicates are shown in the figure in the figure legend.
8) In line 196 it would be helpful to add in parenthesis that Rimonabant is SR 141716A so reader doesn’t have to refer back to the methods to clarify.
9) Figure 3 title is only encompassing panels D and E so it would be appropriate to also include information about Panels A-C-.
10) Please add an appropriate data availability statement.
11) In the simple summary, the authors finish by stating “demonstrate that they help to recruit the immune system to fight the patient’s tumor”. Considering the authors did not do any in-vivo studies that would allow them to measure immune cell recruitment this should be softened by saying “suggesting” or a similar phrase rather than demonstrating.
